# Characterization and expression analysis of genes encoding three small heat shock proteins in the oriental armyworm, *Mythimna separata* (Walker)

**Hong-Bo Li** *, **Chang-Geng Dai, Yang Hu**

Institute of Plant Protection, Guizhou Academy of Agricultural Sciences, Guiyang, China

* gzlhb2017@126.com

**Data Availability Statement:** All relevant data are within the paper and its Supporting Information files.

## Abstract

Small heat shock proteins (sHsps) function in the response of insects to abiotic stress; however, their role in response to biotic stress has been under-investigated. *Mythimna separata*, the oriental armyworm, is polyphenetic and exhibits gregarious and solitary phases in response to high and low population density, respectively. In this study, three genes were identified encoding sHsps, namely *MsHsp19.7*, *MsHsp19.8* and *MsHsp21.4*, and expression levels in solitary and gregarious *M. separata* were compared. The deduced protein sequences of the three *MsHsps* had molecular weights of 19.7, 19.8 and 21.4 kDa, respectively, and contained a conserved α-crystalline domain. Real-time PCR analyses revealed that the three *sHsps* were transcribed in all developmental stages and were dramatically up-regulated at the 6th larval stage in gregarious individuals. Expression of the three *MsHsps* was variable in different tissues of 6th instar larvae, but exhibited consistent up- and down-regulation in the hindgut and Malpighian tubules of gregarious individuals, respectively. In addition, *MsHsp19.7* and *MsHsp19.8* were significantly induced when solitary forms were subjected to crowding for 36 h, but all three *MsHsps* were down-regulated when gregarious forms were isolated. Our findings suggest that population density functions as a stress factor and impacts *MsHsps* expression in *M. separata*.

## Introduction

High population density (crowding) is a complex stress that impacts the morphology, behavior, life history and physiology of insects [1–3] and their population dynamics in the field [4]. To overcome the unfavorable effects of crowding, insects skillfully adopt one or more strategies. For example, insects may alter their phenotype or behavior to adapt to crowding or they may reallocate resources normally used for basic functions (e.g. development, reproduction, nutrient assimilation, and immunity) to cope with changes in population density [5–7]. Phase polyphenism is a phenotypic adaption to crowding that has been observed in Orthopterans, Lepidopterans, Hemipterans and Coleopterans [6, 8–10]. Solitary and gregarious phases have

**Funding:** This researchwas surproted by the National Natural Science Foundation of China (31601633), the National Key Research and Development Programme of China (2018YFD0200700),Talent Foundation of Guizhou Academy of Agricultural Science(201301).

**Competing interests:** The authors have declared that no conflict of interests exist.

been observed in selected species when subjected to low and high population density, respectively. Gregarious individuals are typically characterized by darker or more melanized cuticles than that of solitary forms [11]. Furthermore, variations in morphology, behavior, life history and disease resistance have been reported in the two insect phases [1, 11–15].

Heat shock proteins (Hsps) are biosynthesized in response to a variety of stressors. As molecular chaperones, Hsps perform critical functions in protein folding, assembly, degradation, and intracellular localization under hospitable and inhospitable conditions [16–18]. Insect Hsps can be classified into four general families, e.g. Hsp90, Hsp70, Hsp60, or small Hsps (sHsps); these families are named according to protein size and structural characteristics [19]. sHsp family members exhibit high diversity due to variability in function, structure, and size (12–43 kDa) [20, 21]. sHsps usually prevent protein aggregation and facilitate the correct refolding of denatured proteins under diverse stressful conditions, such as heat, cold, oxidation, drought, UV radiation, hypertonic stress and chemical exposure [19]. Apart from the stress response, some sHsps also function in insect metamorphosis and development [22–25], longevity [26] and diapause [27–30]. Recently, some studies have reported that sHsps are also involved in immune responses when insects are colonized by infectious microorganisms [27, 31]. However, studies on sHsps have largely focused on model insects and sHsp roles in response to abiotic stress, including extreme temperature, UV irradiation, oxidation, chemicals expsoure, etc. Little is known about sHsp functions in response to biotic stressors such as variations in population density.

*Mythimna separata* (Walker), which is commonly known as the oriental armyworm, is a formidable pest in Asia. *M. separata* exhibits polyphenism with solitary and gregarious phases occurring at low and high density, respectively [32], which provides an ideal model to investigate if population density functions as a stressor that impacts organismal physiology [33]. Although the up-regulation of *Hsc70* has been observed in gregarious *M. separata* larvae [34], it is not clear how other Hsps respond to alterations in population density. In this report, we investigate whether the sHsp genes, *MsHsp19.7*, *MsHsp19.8* and *MsHsp21.4*, are up-regulated by alterations in *M. separata* population density and if variability occurs among gregarious and solitary phases. Our findings provide some understanding of the ecological impact of *sHsp* expression in the evolution and adaptation of *M. separata*.

## Materials and methods

### Ethics statement

The *M. separata* larvae were collected from corn stalks cultivated in Qianxi county, Guizhou province (27°01′39.72″N, 106°20′2.92″E), in 2015. In present study, there were no specific permits being required for the insect collection. No endangered or protected species were involved in the field studies. The "List of Protected Animals in China" does not contain the *M. separata* which are common insect.

### Insects

Solitary and gregarious *M. separata* were raised at the Institute of Plant Protection, Guizhou Academic of Agricultural Sciences, China as described previously [32]. Briefly, gregarious cultures were raised in 1 L cylinders (40 neonates/container) and solitary individuals were reared in 300 mL cylinders (1 neonates/container) [32]. Insects were fed on corn leaves and maintained as described [32]. The two phases were raised for five or more generations prior to experiments.

## Preparation of samples

Developmental stages (e.g. eggs, 1st-6th instar larvae, pupae, and adults) and tissues of 6th instar larvae (heads, epidermis, foregut, midgut, hindgut and Malpighian tubules) were collected from solitary and gregarious insects as described [32], frozen in liquid nitrogen, and stored at -80˚C until analysis.

The impact of crowding and isolation were evaluated by crowding solitary forms of *M. separata* and isolating gregarious *M. separata*, respectively. Sixth instar larvae of solitary *M. separata* were subjected to crowding by grouping 40 individuals in a 1 L cylinder; conditions for isolation involved separating 6th instar larvae of gregarious *M. separata* and placing them in individual 300 mL plastic cylinders [32]. Following treatment, the samples were collected and profiles were examined for expression of the three *sHsp* genes. Treatments consisted of three larvae and were replicated three times.

## RNA isolation, cDNA synthesis and RT-PCR

The SV Total RN A isolation system was used to extract total RNA as recommended (Promega, WI, USA), and DNase I was used to remove residual genomic DNA. RNA quality was evaluated by electrophoresis and UV spectrophotometry as described [32]. The First Strand cDNA Synthesis Kit cDNA was used to generate cDNA from 1 µg total RNA as recommended (Fermentas, Canada), and cDNAs were stored at -20˚C until needed.

Degenerate primers were designed according to the conserved α-crystallin domains of sHsps genes from Noctuidae species, and used to amplify partial sequences of three *M.separata* sHsp genes by RT-PCR (S1 Table). The reaction conditions for PCR, extraction from agarose gels, cloning, and sequencing followed established protocols [32].

The obtained partial sequences of the three sHsp genes were utilized to design gene-specific primers. Total RNA (1 µg) was used in 5′- and 3′-RACE with the SMARTer® RACE 5'/3' Kit as recommended (Takara Bio USA, Inc.) (S1 Table). RACE was conducted and PCR products were purified, cloned and sequenced as described previously [32]. The initial cDNA and 5′- and 3′-RACE products were assembled to obtain full-length cDNA.

## Bioinformatic analysis

The open reading frames (ORFs) were detected with ORF Finder (https://www.ncbi.nlm.nih.gov/orffinder/), and sequences were aligned with ClustalW (https://embnet.vital-it.ch/software/BOX_form.html). The predicted mass and isoelectric point for each sHsp were calculated with Compute pI/Mw (https://web.expasy.org/compute_pi/). Conserved motifs were annotated using the NCBI Conserved Domain Database (http://www.ncbi.nlm.nih.gov/Structure/cdd/wrpsb.cgi). Phylogenetic trees were constructed by MEGA v. 7 (https://www.megasoftware.net) using themaximum likelihood method with 2000 bootstrap replicates.

## Quantitative Real-Time PCR (qRT-PCR)

qRT-PCR was executed using a BioRad CFX96 system (Hercules, CA, USA) in a 20 µL reaction volume containing SsoAdvanced Universal SYBR Green Supermix (10 µL, Bio-Rad), gene-specific primers (1 µL each, S1 Table), cDNA template (1 µL), and ddH$_2$O (7 µL). PCR and melting curve analysis were conducted using established parameters [32]. *Actin* was used to normalize transcript abundance for developmental stage samples, and *Tubulin* was used to normalize expression for different tissues and population densities [35]. Every treatment contained four replications, and each replication contained triplicate samples.

## Statistical analysis

Data were expressed as means ± SE. The comparative Ct method was used to calculate relative expression levels and expressed as $2^{-\triangle\triangle Ct}$ [36]. Differences between solitary and gregarious phases were discovered using the Student's t-test, and results were considered significant at $P<0.05$. Data Processing System (DPS) software was used to analyze the results [37].

## Results

### Characterization of three sHsps genes

Three *sHsp* genes were obtained from *M. separata*, and named *MsHsp19.7*, *MsHsp19.8*, and *MsHsp21.4* based one their respective predicted molecular weight (GenBank accession number: MN503276, MN503277, and MN503278, respectively). *MsHsp19.7*, *MsHsp19.8*, and *MsHsp21.4* encoded 528, 534, and 564 bp ORFs with deduced translational products containing 175, 177 and 187 amino acids, respectively. The predicted sizes of MsHsp19.7, MsHsp19.8, and MsHsp21.4 were 19.7, 19.8 and 21.4 kDa with isoelectric points of 6.53, 6.08 and 5.79, respectively. Multiple sequence alignments revealed that the three MsHsps contained a conserved α-crystallin domain, which was composed of approximately 100 amino acids and six β-strands (Fig 1). Blast analysis showed that the deduced amino acids of three sHsps shared a reasonable degree of identity with their respective homologs from other Lepidoptera insects.

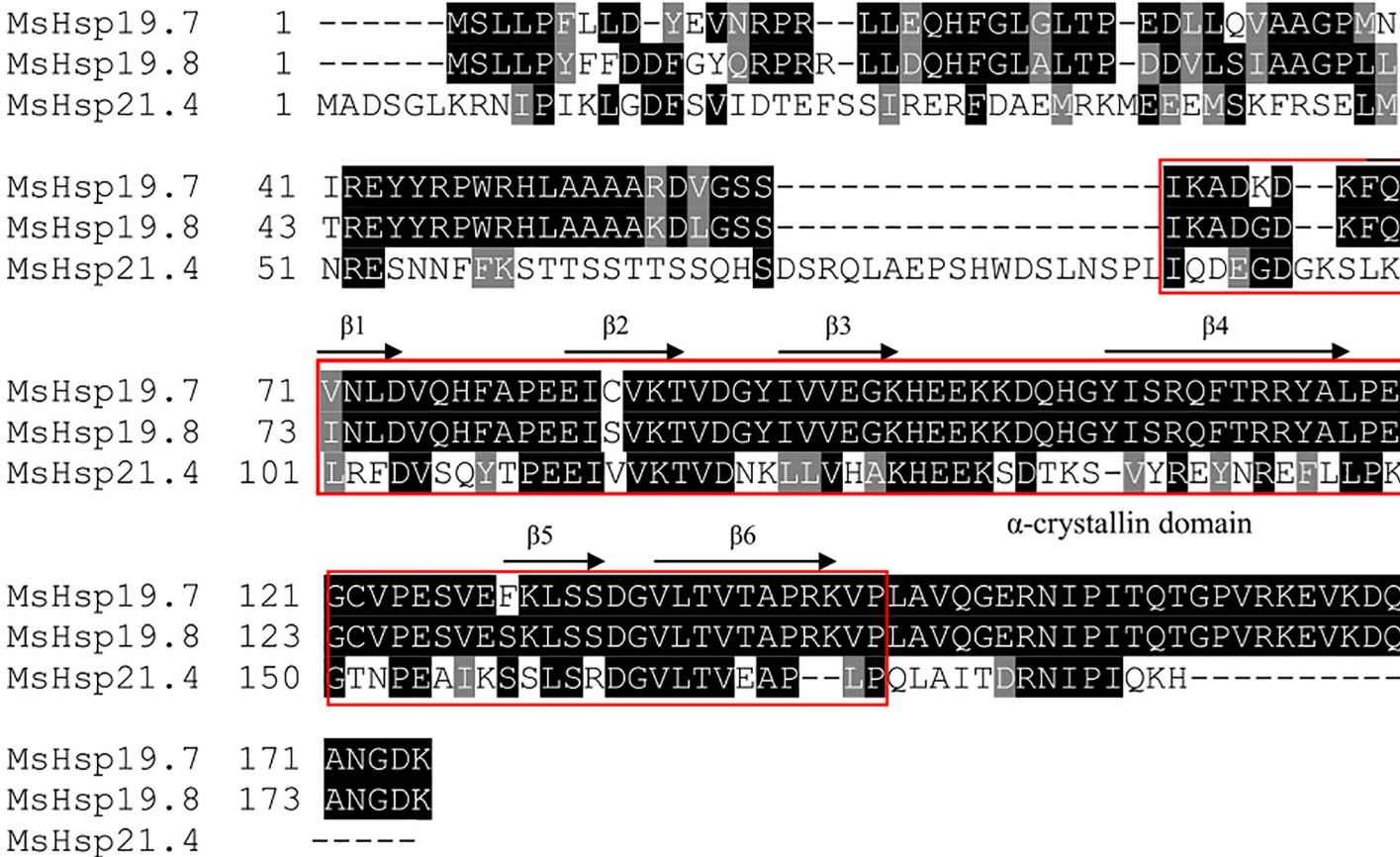

**Fig 1. ClustalW alignment of MsHsp19.7, MsHsp19.8, and MsHsp21.4 from *M. separata*.** The conserved α-crystallin domain is demarcated by a red rectangle. Six β-strands in the α-crystallin domain are indicated by black arrows.

For example, MsHsp19.7 had 90.86% identity with Hsp19.7 in *Helicoverpa armigera*, and MsHsp19.8 showed 84.57% identiy with Hsp20.6 in *Spodoptera litura*, while MsHsp21.4 shared 86.44% similarity with homolog in *Mamestra brassicae*.

## Phylogenetic analysis of the three MsHsps

Twenty two sHsps, including twenty from Lepidopteran species and two from *D.melanogaster*, were downloaded from NCBI and maximum likelihood method was used to generate a phylogenetic tree. As shown in Fig 2, MsHsp19.7 and MsHsp19.8 were assigned to a cluster, and separated from MsHsp21.4. Specifically, MsHsp19.7 grouped with HaHsp19.7(*Helicoverpa*

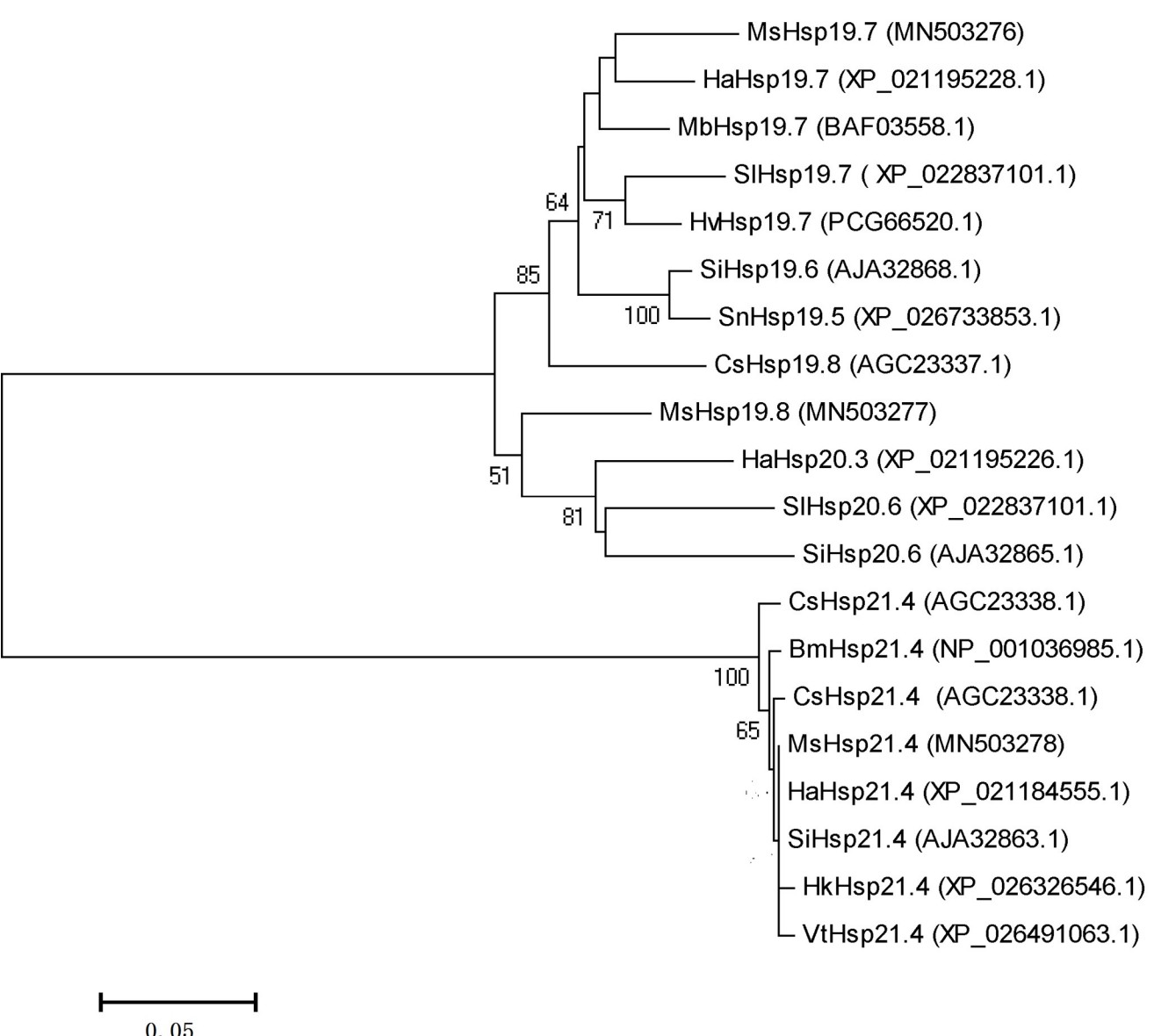

**Fig 2. Phylogenetic analysis of sHsps.** The maximum likelihood algorithm was used to generate a phylogenetic tree based on 22 sHsps, including 20 sHsps from Lepidopteran species and 2 sHsps from *Drosophila melanogaster*(outgroup) (S2 Table). Nodes were labeled with percent bootstrap values from 2000 re-sampling events, and values less than 50 were deleted.

*armigera*), MsHsp19.8 clustered with SlHsp20.6 (*Spodoptera litura*), and MsHsp 21.4 was closely related to HaHsp21.4. These results confirmed that the MsHsp proteins were members of the sHsp family.

## Developmental expression profiles

Transcription of the three *MsHsps* genes in various developmental stages of solitary and gregarious *M. separata* was investigated by qRT-PCR. *MsHsp19.7* expression was not significantly different in solitary and gregarious phases at the $2^{nd}$ larval and adult stages ($2^{nd}$, $t = 2.42$, $P = 0.072$; A, $t = 1.51$, $P = 0.206$); however, significant differences were observed in the other developmental stages (E, $t = 4.48$, $P = 0.011$; $1^{st}$ instar, $t = 4.40$, $P = 0t.012$; $3^{rd}$, $t = 3.24$, $P = 0.032$; $4^{th}$, $t = 4.68$, $P = 0.009$; $5^{th}$, $t = 3.62$, $P = 0.022$; $6^{th}$, $t = 68.113$, $P = 0.001$; P: $t = 2.86$, $P = 0.046$) (Fig 3A). No differences were observed for *MsHsp19.8* expression in the $1^{st}$ and $3^{rd}$ instar larvae ($1^{st}$, $t = 1.23$, $P = 0.286$; $3^{rd}$, $t = 0.20$, $P = 0.849$), but solitary and gregarious phases showed obvious differences in other stages (E, $t = 9.15$, $P = 0.008$; $2^{nd}$, $t = 10.00$, $P = 0.010$; $4^{th}$, $t = 5.51$, $P = 0.005$; $5^{th}$, $t = 6.59$, $P = 0.003$; P: $t = 8.54$, $P = 0.001$; A, $t = 3.40$, $P = 0.02$) (Fig 3B). Significant differences were observed for *MsHsp21.4* expression in the two phases at the $1^{st}$, $4^{th}$, $5^{th}$ and $6^{th}$ larval stages ($1^{st}$, $t = 3.91$, $P = 0.017$; $4^{th}$, $t = 3.64$, $P = 0.022$; $5^{th}$, $t = 5.64$, $P = 0.005$; $t = 4.49$, $P = 0.011$) (Fig 3C). In general, the three *sHsp* genes were more highly expressed in gregarious $6^{th}$ instar larvae as compared to solitary $6^{th}$ instar larvae (Fig 3A–3C).

## Tissue-specific expression profiles

Tissue-specific expression was analyzed in gregarious and solitary forms of $6^{th}$ instar larvae (Fig 4). In HG tissues, the three *MsHsps* were expressed at 3.07–4.17-fold higher levels in gregarious larvae as compared to solitary individuals (*MsHsp19.7*: $t = 5.51$, $P = 0.030$; *Hsp19.8*: $t = 8.60$, $P = 0.001$; *Hsp21.4*: $t = 4.70$, $P = 0.040$). In contrast, the three *MsHsps* were expressed at 1.60–3.28-fold higher levels in the Malpighian tubules (MT) of solitary individuals as compared to gregarious larvae (*MsHsp19.7*: $t = 3.16$, $P = 0.034$; *MsHsp19.8*: $t = 11.09$, $P<0.001$; *MsHsp21.4*: $t = 2.80$, $P = 0.049$). In heads (HD), *MsHsp19.7* and *MsHsp19.8* were more highly expressed in gregarious larvae than solitary ones, but *MsHsp21.4* expression was not significantly different (*MsHsp19.7*: $t = 10.11$, $P = 0.009$; *MsHsp19.8*: $t = 12.37$, $P = 0.006$; *MsHsp21.4*: $t = 0.19$, $P = 0.855$). In epidermis (ED), *MsHsp21.4* expression was higher in gregarious individuals, whereas the other two *sHsps* expression are higher in solitary ones. (ED: *MsHsp19.7*: $t = 4.10$, $P = 0.015$; *MsHsp19.8*: $t = 2.88$, $P = 0.045$; *MsHsp21.4*: $t = 12.42$, $P<0.001$). In MG (midgut) tissues, *MsHsp19.8 and MsHsp21.4* expression were significantly higher in gregarious larvae than solitary ones, but *MsHsp19.7* expression was not different between the two phases (*MsHsp19.7*: $t = 0.24$, $P = 0.819$; *MsHsp19.8*: $t = 5.23$, $P = 0.034$; *MsHsp21.4*: $t = 7.06$, $P = 0.019$). However, the three *MsHsps* showed variable expression in foregut (FG) tissues in the two phases (*MsHsp19.7*: $t = 1.52$, $P = 0.202$; *MsHsp19.8*: $t = 4.46$, $P = 0.009$; *MsHsp21.4*: $t = 12.67$, $P<0.001$).

## Isolation and crowding-induced expression profiles

Expression levels of the three *MsHsps* were significantly impacted by alterations in population density. Expression of *MsHsp19.7* and *MsHsp19.8* were upregulated when solitary *M. separata* larvae were subjected to crowding for 36 h; however, expression of *MsHsp21.4* was not significantly changed (*MsHsp19.7*: $t = 3.71$, $P = 0.021$; *MsHsp19.8*: $t = 3.36$, $P = 0.032$; *MsHsp21.4*: $t = 1.06$, $P = 0.350$) (Fig 5A, 5C and 5E). In contrast, expression levels of the three *MsHsps* were down-regulated in gregarious individuals subjected to isolation for 36 h (*MsHsp19.7*: $t = 12.03$, $P<0.001$; *MsHsp19.8*: $t = 5.14$, $P = 0.011$; *MsHsp21.4*: $t = 18.42$, $P<0.001$) (Fig 5B, 5D and 5F).

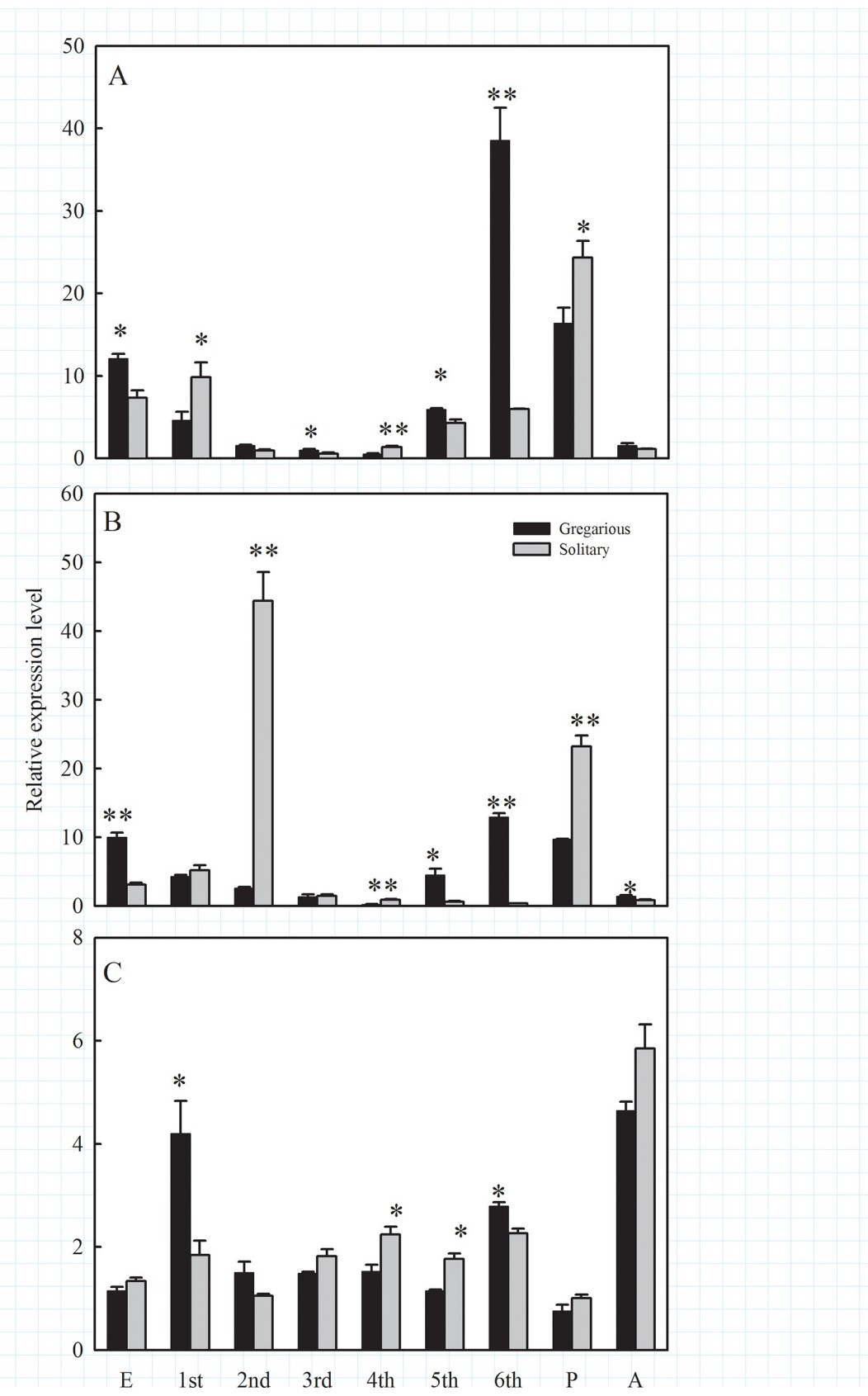

**Fig 3. Expression of *MsHsps* in solitary and gregarious *M. separata*.** Panel (A) *MsHsp19.7*; (B), *MsHsp19.8*; and (C) *MsHsp21.4*. Abbreviations: E, eggs; 1st, 2nd, 3rd, 4th, 5th and 6th, first through sixth instar larvae; P, pupae; and A, adults. Data points represent means ($n = 3$) and error bars denote standard deviation (SD). Significant differences (Student's *t*-test) between the two phases of *M. separata* are shown with asterisks (*, $P<0.05$; **, $P<0.01$).

## Discussion

In this study, three genes encoding sHsps (*MsHsp19.7*, *MsHsp19.8* and *MsHsp21.4*) were identified in *M. separata*. Nucleotide sequencing revealed that the three *MsHsps* encoded proteins with similarity to sHsps reported in Noctuidae species and contained the conserved α-crystalline domain that has been reported previously [21, 23, 24]. Phylogenetic analysis of sHsps indicated that MsHsp19.7 and MsHsp21.4 are related to respective orthologs in other Lepidopteran species. An exception was MsHsp19.8, which clustered with Hsp20.6 orthologs; in this context, our findings are analogous to those observed for *Chilo suppressalis* [24], *Choristoneura fumiferana* [30], *Spodoptera litura* [23], *Bombyx mori* [38] and *Grapholitha molesta* [39]. This level of phylogenetic diversity may be caused by different rates of sHsp evolution and/or the functional diversity of sHsps in insects.

sHsps are known for regulating insect development. In this study, the three *MsHsps* were expressed in all developmental stages of solitary and gregarious *M. separata*, suggesting their importance throughout the *M. separata* lifespan. Variability in life history, morphology, and behavior has been detected in solitary and gregarious forms of *M. separata* [40, 41]. Contrary to expectation, expression of the three *MsHsps* was not consistently higher in gregarious individuals from egg to the 5th larval stage, which was similar to results obtained with locusts [33]. A possible explanation is that the small body size evident in these stages reduces contact between individual insects, thus alleviating the crowding-induced stress response [42]. However, in the 6th instar larvae of gregarious individuals, expression of the three *MsHsps* was significantly upregulated, potentially due to the increased competition for resources [32]. Interestingly, up-regulation of the three *MsHsps* was not observed in gregarious pupae or adults, possibly because this pest undergoes dramatic changes in metamorphosis at these stages and crowding becomes less critical.

The three *MsHsps* were expressed in tissues of both phases, but showed tissue-specific expression patterns, thus suggesting that MsHsps contribute to normal functioning of the organism [43]. Specifically, expression of *MsHsp19.7* and *MsHsp19.8* in heads was higher in gregarious versus solitary larvae, suggesting that these two MsHsps may respond to the stress signal(s) produced during crowding. The three *MsHsps* were also upregulated in the hindgut of gregarious larvae, but showed lower levels of expression in Malpighian tubules (Fig 5). Malpighian tubules and hindgut are known to reabsorb water, salts, and other substances before excretion by the insect [24]. Previous studies have shown that gregarious larvae have higher food consumption than solitary forms [40], which could lead to higher production of toxic by-products. Therefore, the higher expression levels of *MsHsps* in gregarious *M. separata* may be needed to protect the hindgut from injury. It remains unclear why expression of the three *MsHsps* are upregulated in Malpighian tubules of gregarious *M. separata*, and further studies are need to address this observation.

Previous reports revealed that alterations in population density could induce *Hsp* expression [2, 32, 44, 45]. In this study, a significant down-regulation of the three *MsHsps* was observed in gregarious *M. separata* exposed to isolation for 36 h; therefore, a reduction in crowding-induced stress in *M. separata* larvae correlated with a decline in population density. In contrast, a dramatic upregulation of *MsHsp19.7* and *MsHsp19.8* was observed in solitary *M. separata* after crowding for 36 h, which was similar to reported studies in locusts [33, 45].

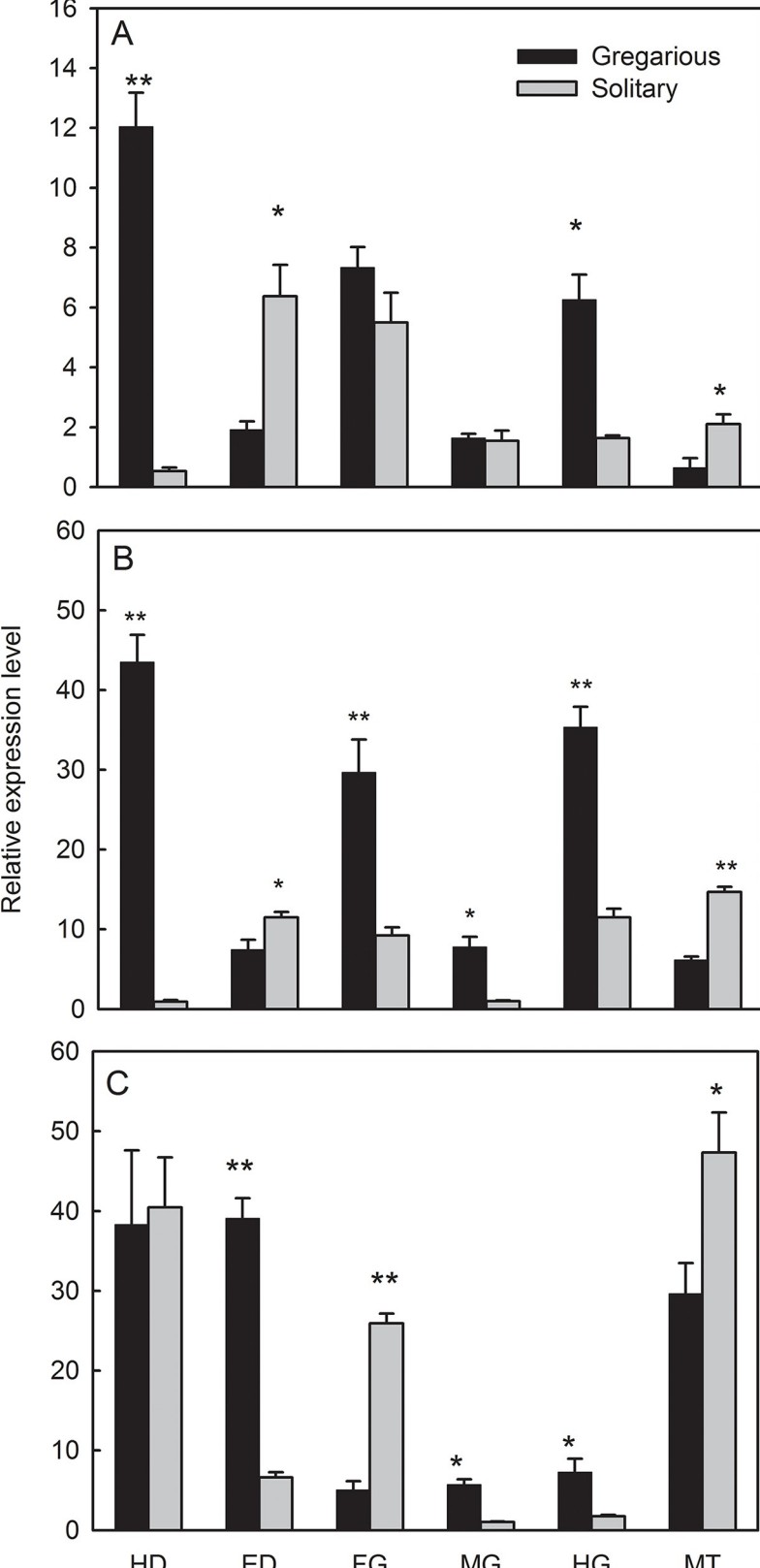

**Fig 4.** Tissue-specific expression profiles of *MsHsp19.7* (A), *MsHsp19.8* (B) and *MsHsp21.4* (C) in 6th instar larvae of solitary and gregarious *M. separata*. Abbreviations: HD, head; ED, epidermis; FG, foregut; MG, midgut; HG, hindgut; and MT, Malpighian tubules. Data points represent mean values ($n$ = 4) with error bars showing SD. Asterisks indicate significant differences between the solitary and gregarious phases of *M. separata* at $P < 0.05$ (*) and $P < 0.01$ (**).

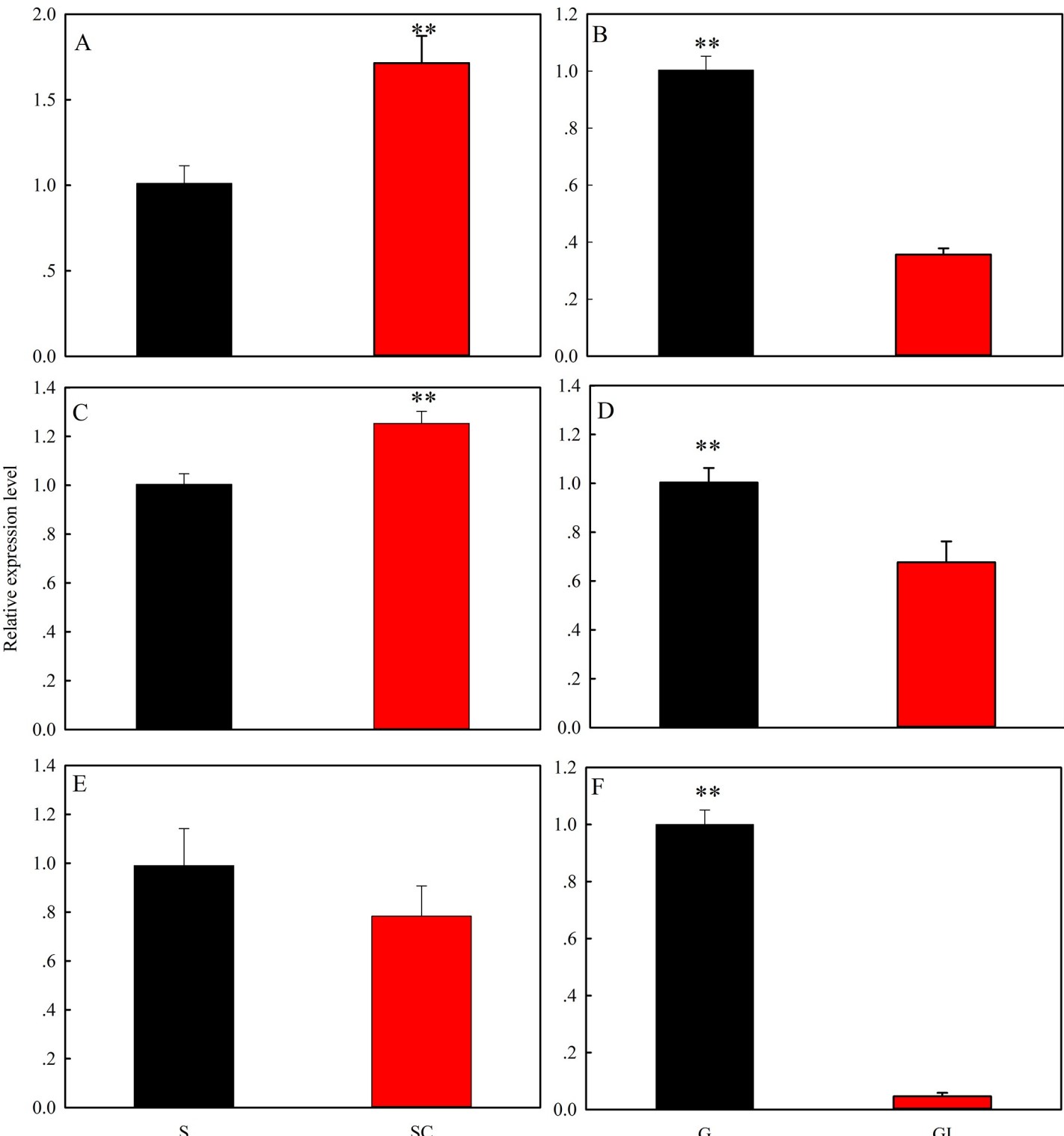

**Fig 5. Expression profiles of *MsHsp* genes in 6ᵗʰ instar larvae subjected to crowding and isolation.** Panels A, C and E show expression of *MsHsp19.7*, *MsHsp19.8* and *MsHsp21.4* in solitary *M. separata* larvae exposed to crowding, respectively. Panels B, D and F show *MsHsp19.7*, *MsHsp19.8* and *MsHsp21.4* expression levels in gregarious *M. separata* larvae subjected to isolation, respectively. Abbreviations: S, solitary; SC-36, solitary larvae exposed to crowding for 36 h; G, gregarious; and GI-36, gregarious larvae exposed to isolation for 36 h. Data points represent mean values (*n* = 4) with error bars denoting SD. Asterisks indicate significance at *P*≤0.01 based on the Student's *t*-test.

However, the expression of *MsHsp21.4* remained unchanged, suggesting that this gene was not induced and/or a longer period may be needed for crowding-induced changes in transcription. Interestingly, recent studies have showed that crowding resulted in down-regulation of *sHsps* in *Drosophila* [2], suggesting that *sHsp* transcription can vary with the organism and its unique response to changes in population density [32].

Prior investigations demonstrated that the upregulation of *Hsps* had negative physiological impacts [18, 46, 47]. Gregarious *M. separata* generally have smaller body sizes and reduced reproduction as compared to solitary individuals [40, 41]. It has been reported that upregulation of small heat shock proteins enhanced resistance to stress in *Locusta* and *Drosophila*, but this was accompanied by a decline in reproduction [48, 49]. Therefore, a trade-off exists between sHsp production and pupal size and reproduction during crowding. In addition, a faster developmental rate has been also observed in gregarious individuals [40, 41], which was likely an environmental adaption to crowding.

Recent reports indicate that gregarious larvae have developed resistance to selected biopesticides [50], which is associated with improved immune system functionality [41, 51]. In insects, sHsps play an important role in the immune response [27, 31]. Therefore, these studies promote our hypothesis that the upregulation of sHsps in gregarious *M. separata* may contribute to improved resistance to biopesticides and pathogens. Newly developed technologies, such as RNAi and CRISPR-Cas9 are needed to confirm this hypothesis in future studies. The three *MsHsps* identified herein may ultimately provide new molecular targets for managing *M. separata* during crowding.

## Conclusion

In summary, three genes encoding small heat shock protein (*sHsps*) were successfully characterized in *M. separata*. Expression analysis by qRT-PCR showed that the three *MsHsps* exhibited variable expression profiles in gregarious and solitary individuals. Moreover, alterations in population density caused large changes in *MsHsp* expression. Our findings show that *MsHsps* function in stress-induced changes that arise due to variations in population density. These findings provide valuable information on the roles of *MsHsps* in *M. separata* populations undergoing fluctuations in population density.

## Supporting information

**S1 Table.**
(DOC)

**S2 Table.**
(DOCX)

## Acknowledgments

We thank Carol L.Bender for language edition.

## Author Contributions

**Conceptualization:** Hong-Bo Li, Yang Hu.

**Data curation:** Hong-Bo Li.

**Funding acquisition:** Hong-Bo Li.

**Investigation:** Hong-Bo Li, Chang-Geng Dai.

Project administration: Hong-Bo Li.

Software: Hong-Bo Li.

Supervision: Hong-Bo Li.

Validation: Hong-Bo Li.

Visualization: Hong-Bo Li.

Writing – original draft: Hong-Bo Li.

Writing – review & editing: Yang Hu.

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
