## [Decision Letter · Decision Letter 0]

25 May 2020

PONE-D-20-04490

Characterization and expression analysis of genes encoding three small heat shock
proteins in the oriental armyworm, Mythimna separata (Walker)

PLOS ONE

Dear Dr. Li,

Thank you for submitting your manuscript to PLOS ONE. After careful consideration, we
feel that it has merit but does not fully meet PLOS ONE’s publication criteria as it
currently stands. Therefore, we invite you to submit a revised version of the
manuscript that addresses the points raised during the review process. Please see
the Reviewer comments appended at the bottom of this email. In addition, I ask that
you consider my comments below.

Editor comments -

Pg3 Line 5 - the Morrow citation is different from the other citations

Pg5 LIne 3 - provide more details on how the degenerate primers were designed. Were
they designed to a specific region? If so, indicate what that region is. Were they
based on previously published primers? If so, provide the appropriate reference.

Pg 5 Line 19 - Perhaps replicate with a more robust analysis such as maximum
likelihood and include an outgroup to root the tree.

Pg 6 LIne 6 - Were amplification efficiencies determined and factored into the
analysis?
It
is widely accepted that studies presenting qRT-PCR data should follow the MIQE
guidelines (see Bustin et al. 2009; Clinical Chemistry 55:611–622).

Pg 6 Line 18 - Indicate that the respective sHsps were named/defined based on the
predicted MW.

Pg 6 - Was BLAST analysis of the predicted proteins done? If so, a Table or
Supplementary Table could be included to indicate the top hits.

Pg 7 Line 7 - the phylogenetic results support their annotation as sHsps they do not
confirm this

Pg 8 Line 11 states that the three sHsps had similar expression patterns in epidermis
regardless of the phase. However, the data shown in figure 4 indicate that sHsp21.4
is higher in gregarious, whereas the other two sHsps are higher in solitary
epidermis. Please clarify.

Pg 11 Line 14 - the proposal that the sHsps contribute to improved resistance is
highly speculative. Unless there is prior data (which should be cited and explicitly
described) indicating that gregarious phase M. separata support this assertion, then
this would be better described as a hypothesis worth testing in future studies.

---

Please submit your revised manuscript by Jul 09 2020 11:59PM. If you will need more
time than this to complete your revisions, please reply to this message or contact
the journal office at plosone@plos.org. When
you're ready to submit your revision, log on to https://www.editorialmanager.com/pone/ and select the 'Submissions
Needing Revision' folder to locate your manuscript file.

If you would like to make changes to your financial disclosure, please include your
updated statement in your cover letter. Guidelines for resubmitting your figure
files are available below the reviewer comments at the end of this letter.

We look forward to receiving your revised manuscript.

Kind regards,

J Joe Hull, Ph.D.

Academic Editor

PLOS ONE

Journal Requirements:

Reviewers' comments:

Reviewer's Responses to Questions

**Comments to the Author**

1. Is the manuscript technically sound, and do the data support the conclusions?

Reviewer #1: Yes

Reviewer #2: Partly

2. Has the statistical analysis been performed
appropriately and rigorously? 

Reviewer #1: Yes

Reviewer #2: Yes

3. Have the authors made all data underlying the
findings in their manuscript fully available?

Reviewer #1: Yes

Reviewer #2: Yes

4. Is the manuscript presented in an intelligible
fashion and written in standard English?

Reviewer #1: Yes

Reviewer #2: Yes

5. Review Comments to the Author

Reviewer #1: This study investigated three small Hsps in the species the oriental
armyworm. The authors cloned the three genes, analyzed the sequences and performed
expression analysis in different developmental stages and tissues as well as in
different population density. The paper provides useful information on the expresson
of the three small Hsps in response to developmental and density changes. I have the
following concerns that needs be addressed by the authors:

1. In Fig. 1, α-crystallin domain should be conserved. However, as shown in this
figure, Hsp21.4 presents non-conseved sequence with the other two small Hsps. Please
clarify it by comparing with the conserved crystallin domain.

2. The relative mRNA levels of small Hsps between gregarious and solitarious forms
varied with developmental stages and tissues. The authors should discuss the results
and give explanations.

3. The following sentence in Abstract is not exactly stated: "Real-time PCR analyses
revealed that the three sHsps were transcribed in all developmental stages and were
dramatically up-regulated at the 6th larval stage in gregarious individuals." Only
hsp19.7 but not other small hsps is dramatically upregulated at the 6th stage.

Reviewer #2: I am confused with the key point. Whether population density induced Hsp
expression or Hsp up/down expression induced gregarious and solitary phases? Rooting
a phylogenetic tree with an outgroup organism or sequence will be better. And latest
version of Mega, too. By the way, consecutive numbering will be better for a
reviewer.

Please see below:

Page 2

Line 14: change “Orthopteran, Lepidopteran, Hemipteran and Coleopteran species” to
“orthopterans, lepidopterans, hemipterans and coleopterans”

Line 17: change “than solitary forms” to” than that of solitary forms”

Line 23: one of, delete

Page 3

Line 7: Unfortunately, delete

Line 8: abiotic stress, explain in detail

Lines 12-14, 16-18: reorganize the sentence

Page 4

Line 3: solitary individuals, how many?

Line 4: as described, explain in detail

Lines 8-10: sample size

Line 20: PCR primers

Page 5

Line 14: “ORF”, what’s this?

Line 19: “MEGA v. 5”, why “v. 5”, why not use the latest version?

Page 6

Lines 1-3: total volume is not 20 μL.

Line 13-14: DPS software, which procedure?

Line 23: “isoelectric points”, predicted isoelectric points

Page 7

Line 1: rooting a phylogenetic tree with an outgroup organism or sequence will be
better. How to build the phylogenetic tree should be mentioned in detail in the
methods section.

Lines 6-8: “HaHsp”, what’s this?

Line 12: the 2nd larvae looks different, why? Explain more in discussion.

Page 9

Line 6: “in Noctuidae species”, in other Noctuidae species

Page 10

Lines 3-15: why these tissues? Which one is the key tissue? And why?

6. PLOS authors have the option to publish the peer
review history of their article (what does this mean?). If published, this will
include your full peer review and any attached files.

If you choose “no”, your identity will remain anonymous but your review may still be
made public.

**Do you want your identity to be public for this peer review?** For
information about this choice, including consent withdrawal, please see our
Privacy Policy.

Reviewer #1: No

Reviewer #2: No

---

## [Author Response · Author response to Decision Letter 0]

19 Jun 2020

Dear editor:

According to the comments from you and two reviewers, we revised our manuscripts
carefully and made some mofications. More details are as follows:

Editor comments:

1. Pg3 Line 5 - the Morrow citation is different from the other citations

Response: We change this citation form according to requirement of Plos one

2. Pg5 LIne 3 - provide more details on how the degenerate primers were designed.
Were they designed to a specific region? If so, indicate what that region is. Were
they based on previously published primers? If so, provide the appropriate
reference.

Response: We described the degenerate primers information in detail, please see page
5 line3-5

3. Pg 5 Line 19 - Perhaps replicate with a more robust analysis such as maximum
likelihood and include an outgroup to root the tree.

Response: Phylogenetic trees were re-constructed by MEGA v. 7 using the maximum
likelihood method with 2000 bootstrap replicates and the sHsps of D.melanogaste were
used as outgroup.

4. Pg 6 LIne 6 - Were amplification efficiencies determined and factored into the
analysis?It is widely accepted that studies presenting qRT-PCR data should follow
the MIQE guidelines (see Bustin et al. 2009; Clinical Chemistry 55:611–622).

Response: The amplification efficiencies were determined and factored into the
analysis.

5. Pg 6 Line 18 - Indicate that the respective sHsps were named/defined based on the
predicted MW.

Response: We indicated the respective sHsps were named/defined based on the predicted
MW.

6. Pg 6 - Was BLAST analysis of the predicted proteins done? If so, a Table or
Supplementary Table could be included to indicate the top hits.

Response: We conducted BLAST analysis of the predicted proteins , and indicated the
thop hits , please see page page6 line1-6.

7. Pg 7 Line 7 - the phylogenetic results support their annotation as sHsps they do
not confirm this.

Response: We confirmed this restults.

8. Pg 8 Line 11 states that the three sHsps had similar expression patterns in
epidermis regardless of the phase. However, the data shown in figure 4 indicate that
sHsp21.4 is higher in gregarious, whereas the other two sHsps are higher in solitary
epidermis. Please clarify.

Response: We clarify the expresson patterns of three sHsps, please see in page8
line17-22.

9. Pg 11 Line 14 - the proposal that the sHsps contribute to improved resistance is
highly speculative. Unless there is prior data (which should be cited and explicitly
described) indicating that gregarious phase M. separata support this assertion, then
this would be better described as a hypothesis worth testing in future studies.

Response: We described it as a hypothesis worth testing in future studies

Reviewer #1: 

This study investigated three small Hsps in the species the oriental armyworm. The
authors cloned the three genes, analyzed the sequences and performed expression
analysis in different developmental stages and tissues as well as in different
population density. The paper provides useful information on the expresson of the
three small Hsps in response to developmental and density changes. I have the
following concerns that needs be addressed by the authors:

1. In Fig. 1, α-crystallin domain should be conserved. However, as shown in this
figure, Hsp21.4 presents non-conseved sequence with the other two small Hsps. Please
clarify it by comparing with the conserved crystallin domain.

Response: We clarified the α-crystallin domainof three sHsps by multiple comparion. 

2. The relative mRNA levels of small Hsps between gregarious and solitarious forms
varied with developmental stages and tissues. The authors should discuss the results
and give explanations.

Response: In fact, in our original manuscript,we have discussed the results and give
possible reasons,more details in page page 9-10.

3. The following sentence in Abstract is not exactly stated: "Real-time PCR analyses
revealed that the three sHsps were transcribed in all developmental stages and were
dramatically up-regulated at the 6th larval stage in gregarious individuals." Only
hsp19.7 but not other small hsps is dramatically upregulated at the 6th stage.

Response: we cheked the expression patterns throughout the developmental stage, all
three hsps were dramatically up-regulated at the 6th larval stage in gregarious
individuals compared to that of solitary ones.

Reviewer #2: 

I am confused with the key point. Whether population density induced Hsp expression
or Hsp up/down expression induced gregarious and solitary phases? Rooting a
phylogenetic tree with an outgroup organism or sequence will be better. And latest
version of Mega, too. By the way, consecutive numbering will be better for a
reviewer.

Page 2

Line 14: change “Orthopteran, Lepidopteran, Hemipteran and Coleopteran species” to
“orthopterans, lepidopterans, hemipterans and coleopterans”

Response: We changed “Orthopteran, Lepidopteran, Hemipteran and Coleopteran species”
to “orthopterans, lepidopterans, hemipterans and coleopterans”

Line 17: change “than solitary forms” to” than that of solitary forms”

Response: We added “that of”

Line 23: one of, delete

Response: “One of” was deleted

Page 3

Line 7: Unfortunately, delete

Response: “Unfortunately” was deleted.

Line 8: abiotic stress, explain in detail

Response: We give the abiotic stress in detail, including extreme temperature, UV
irradiation, oxidation, chemicals expsoure, etc.

Lines 12-14, 16-18: reorganize the sentence

Response: We reorganized this sentence.

Page 4

Line 3: solitary individuals, how many?

Response: The number of solitary individuals is 1 neonates/container, and the volume
of the container is 300 mL.

Line 4: as described, explain in detail

Response: the raised method was given in our published paper (Li et al., 2019, J 

Econ Entomol), so we have cited this reference. 

Lines 8-10: sample size

Response: the sample size was the same as our published paper(Li et al., 2019. Plos
one), we have cited this reference 

Line 20: PCR primers

Response: We have give the primers.

Page 5

Line 14: “ORF”, what’s this?

Response: “ORF” is the abbreviation of “ open reading frames”.

Line 19: “MEGA v. 5”, why “v. 5”, why not use the latest version?

Response: We used MEAG v.7 to generate the phylogenetic tree

Page 6

Lines 1-3: total volume is not 20 μL.

Response: we checked the composition(SsoAdvanced Universal SYBR Green Supermix (10
μL, Bio-Rad), gene-specific primers (1 μL each, Table S1), cDNA template (1 μL), and
ddH2O (7 μL)), and the volume was 20μL.

Line 13-14: DPS software, which procedure?

Response: We added the procedure( Scientific Press, China) in the methods.

Line 23: “isoelectric points”, predicted isoelectric points

Response: we added the word”predicted”.

Page 7

Line 1: Rooting a phylogenetic tree with an outgroup organism or sequence will be
better. How to build the phylogenetic tree should be mentioned in detail in the
methods section.

Response: We give the procedure of building the phylogenetic tree in detaile, please
see page5 line20-21.

Lines 6-8: “HaHsp”, what’s this?

Response: Hsps in Helicoverpa armigera.

Line 12: the 2nd larvae looks different, why? Explain more in discussion.

Response: Expression of three sHsps failed to show any consistent trend before the
fifth-instar larvae. We think that a possible explanation for such expression is
that the body size in these developmental stages is so small that each individual
may occupy

enough space to lower the mean contact rate between individuals

Page 9

Line 6: “in Noctuidae species”, in other Noctuidae species

Response: We added the word”other”

Page 10

Lines 3-15: why these tissues? Which one is the key tissue? And why? 

Response: We think these tissues include the most tissues. To our knowledge, we think
head, epidemics and guts are key tissues. First, the head is the most important
tissue/organs in sensing environmental stress signals. Secondly, the epidemics is an
important tissue when the gregarious ones contact under high density conditions.
Thirdly, the guts play impornt role in absorbing food and protecting from
injury.

---

## [Editor Report · Decision Letter 1]

25 Jun 2020

Characterization and expression analysis of genes encoding three small heat shock
proteins in the oriental armyworm, Mythimna separata (Walker)

PONE-D-20-04490R1

Dear Dr. Li,

We’re pleased to inform you that your manuscript has been judged scientifically
suitable for publication and will be formally accepted for publication once it meets
all outstanding technical requirements.

Kind regards,

J Joe Hull, Ph.D.

Academic Editor

PLOS ONE
---

## [Editor Report · Acceptance letter]

16 Jul 2020

PONE-D-20-04490R1 

Characterization and expression analysis of genes encoding three small heat shock
proteins in the oriental armyworm, *Mythimna separata* (Walker) 

Dear Dr. Li:

I'm pleased to inform you that your manuscript has been deemed suitable for
publication in PLOS ONE. Congratulations! Your manuscript is now with our production
department. 

Kind regards, 

on behalf of

Dr. J Joe Hull 

Academic Editor

PLOS ONE